# Relationship between Physicochemical Characteristics and Pathogenic *Leptospira* in Urban Slum Waters

**DOI:** 10.3390/tropicalmed5030146

**Published:** 2020-09-16

**Authors:** Daiana de Oliveira, Vladimir Airam Querino, Yeonsoo Sara Lee, Marcelo Cunha, Nivison Nery, Louisa Wessels Perelo, Juan Carlos Rossi Alva, Albert I. Ko, Mitermayer G. Reis, Arnau Casanovas-Massana, Federico Costa

**Affiliations:** 1Instituto Gonçalo Moniz, Fundação Oswaldo Cruz, Ministério da Saúde, Rua Waldemar Falcão, 121, Candeal, Salvador CEP 40296-710, Bahia, Brazil; daisoliveira@hotmail.com (D.d.O.); nivison.nery@conveniado.bahia.fiocruz.br (N.N.J.); albert.ko@yale.edu (A.I.K.); mitermayer.reis@fiocruz.br (M.G.R.); 2Department of Environmental Engineering, Escola Politécnica, Universidade Federal da Bahia, Rua Aristides Novis, 02, Federação, Salvador CEP 40210-630, Bahia, Brazil; vlaairam@yahoo.com.br (V.A.Q.); louperelo@gmail.com (L.W.P.); 3Mayo Clinic Alix School of Medicine, Mayo Clinic, 13400 E. Shea Blvd, Scottsdale, AZ 85259, USA; ysaralee@gmail.com; 4Instituto da Saúde Coletiva, Universidade Federal da Bahia, Rua Basílio da Gama, 316, Canela, Salvador CEP 40110-040, Bahia, Brazil; 5National School of Public Health, Fundação Oswaldo Cruz, Rua Leopoldo Bulhões, 1480-Manguinhos, Rio de Janeiro 05407-002, RJ, Brazil; cunha.ensp@gmail.com; 6Universidade Católica do Salvador, Av. Prof. Pinto de Aguiar, 2589-Pituaçu, Salvador CEP 41740-090, Bahia, Brazil; jcrossiconsult@gmail.com; 7Department of Epidemiology of Microbial Diseases, Yale School of Public Health, 60 College St, New Haven, CT 06510, USA; 8Faculdade de Medicina, Universidade Federal da Bahia, Praça XV de novembro, s/n-Largo do Terreiro de Jesus, Salvador CEP 40026-010, Bahia, Brazil

**Keywords:** leptospirosis, environmental, sewer, standing, LipL32, pH, salinity, TDS

## Abstract

Leptospirosis, a zoonosis caused by pathogenic *Leptospira*, primarily affects tropical, developing regions, especially communities without adequate sanitation. Outbreaks of leptospirosis have been linked with the presence of pathogenic *Leptospira* in water. In this study, we measured the physicochemical characteristics (temperature, pH, salinity, turbidity, electrical conductivity, and total dissolved solids (TDS)) of surface waters from an urban slum in Salvador, Brazil, and analyzed their associations with the presence and concentration of pathogenic *Leptospira* reported previously. We built logistic and linear regression models to determine the strength of association between physicochemical parameters and the presence and concentration of *Leptospira*. We found that salinity, TDS, pH, and type of water were strongly associated with the presence of *Leptospira*. In contrast, only pH was associated with the concentration of the pathogen in water. The study of physico-chemical markers can contribute to a better understanding of the occurrence of *Leptospira* in water and to the identification of sources of risk in urban slum environments.

## 1. Introduction

Leptospirosis is a zoonotic disease with global distribution that has emerged as a health problem in urban slum communities in tropical and developing countries [1], causing over 1 million cases and almost 60,000 deaths every year [2]. The disease includes a range of manifestations: from a sub-clinical illness or mild influenza-like symptoms to severe complications, such as Weil’s disease and pulmonary hemorrhagic syndrome, for which mortality rates are 10% and 50%, respectively [3]. Leptospirosis is caused by pathogenic spirochetes of the genus *Leptospira*. Pathogenic *Leptospira* colonize the kidneys of domestic and wild mammals, notably in rodents who act as chronic carriers, and shed the pathogen in urine [3]. Human infection occurs through contact with infected animal urine or soil and water contaminated with pathogenic *Leptospira* [4]. Therefore, the environment plays a central role in the spillover infections from animals to humans.

Outbreaks of leptospirosis and endemic transmission have been linked with the presence of pathogenic *Leptospira* in water [5]. Consequently, research on the environmental phase of the pathogen is of epidemiological importance, especially to unravel the dynamics of transmission. Physicochemical variables such as temperature, salinity, turbidity, and total dissolved solids (TDS) have been associated with *Leptospira* presence in aquatic environments and survival under laboratory-controlled conditions [6,7,8,9,10]. In general, slightly alkaline waters and high turbidity favor the survival of the bacterium [9,10]. However, little is known about the relationship between water physico-chemical variables and *Leptospira* under field conditions. Specifically, there is a lack of information from waters from urban slums, where leptospirosis is endemic and large outbreaks are reported annually [11,12,13]. In these settings, the lack of sanitary infrastructure and high rates of rat infestation increase human exposure to contaminated environments [14]. In this study, we aimed (i) to investigate the physicochemical parameters of the surface waters (sewage and standing water) from an urban slum in Brazil and (ii) to evaluate the association between these physico-chemical parameters and type of water and the presence and concentration of pathogenic *Leptospira*.

## 2. Materials and Methods

This study was performed in Pau da Lima, Salvador, Brazil, a well characterized urban slum community, with precarious housing, poverty, high population density, trash accumulation, and lack of sanitary and sewer infrastructure. A study previously conducted in this neighborhood revealed high leptospirosis infection rates, with 37.8 cases per 1000 inhabitants [15].

We analyzed sewage and standing water samples collected in a previous study which aimed to characterize the spatio-temporal distribution of pathogenic *Leptospira* [16] in order to evaluate associations with physicochemical characteristics. Samples were collected from 14 sampling sites distributed across an open sewer that flows from the top to the bottom of one of the valleys. Four samples were collected at each site, two from the open sewer and two from standing water located close to the sewer. Sewage was defined as a continuous and open channel that carries household wastewater and rain. Standing water was defined as any body of water accumulated at ground level, with no connection to sewage or other running water, located at up to 10 m from the sewer. Samples were collected in the morning and afternoon of the same day, for three non-consecutive days during one week in July 2011 and one week in January 2012. Duplicate aliquots of 50 mL of sewage and standing water were collected in sterile containers using aseptic techniques and processed within 6 h after collection. One aliquot was used to determine the concentration of *Leptospira* DNA using a Real-time quantitative-PCR (qPCR) targeting the lipL32 gene. The methods and results for the molecular quantification were published previously [16]. The second aliquot was used to measure physiochemical parameters following standard procedures [17]. Temperature was measured in situ before sample collection. pH was determined with a digital pH meter (inoLab) and turbidity using a spectrophotometer (Hach Lange). Salinity, electric conductivity, and total dissolved solids (TDS) were measured using a multiparameter probe (Tetracon 325). The physico-chemical parameters measured here were combined with the previously reported results on *Leptospira* concentration [16] for subsequent statistical analyses and modeling in combination.

Averages and standard deviations of physicochemical variables (temperature, pH, turbidity, TDS, salinity, and electric conductivity) were calculated for qPCR positive and negative samples and stratified by water type. The geometric mean and range of *Leptospira* concentrations were calculated for positive samples. Multiple logistic and linear regressions were used to investigate the relationship between water physicochemical parameters and the presence and concentration of pathogenic *Leptospira*. The results obtained for each sample were grouped by type of sample (sewage or standing water) for the regression analyses. To select the best model, physicochemical variables were iteratively included to examine individual effects. The parameters that presented a value of *p* < 0.20 were included in the multivariate model. The best models were selected based on the lowest Akaike Information Criterion (AIC) value. The final adjusted logistic and linear models accounted for multicollinearity and interaction between variables. All analyses were performed using RStudio v 1.2.5033 [18].

## 3. Results

A total of 284 water samples (166 samples of sewage and 118 standing water) were collected (Appendix A). Mean temperature, pH, TDS, electrical conductivity, and salinity were different between water samples that contained pathogenic *Leptospira* and negative water samples (Table 1). Turbidity was the only parameter that did not show a significant difference between positive and negative water samples. Average pH was 7.1 for positive water samples versus 7.3 negative water samples (*p* = 0.02). Mean turbidity was slightly higher in positive samples, while temperature, TDS, salinity, and water conductivity were higher in negative samples (*p* < 0.05) (Table 1). In an additional analysis (Appendix A), we observed that there was a significant difference in the all physicochemical characteristics by water type (*p* < 0.05). Mean pH was 7.3 in sewage samples versus 7.1 in standing water samples. Mean TDS and salinity were higher in sewage samples than in standing water samples (652 versus 359 mg/L and 0.37 versus 0.12‰, respectively).

The analysis of the parameters included in the final logistic and linear models (odds ratios and coefficients, respectively) is presented in Table 2. The univariate logistic model for the presence of pathogenic *Leptospira* showed that temperature, pH, TDS, salinity, and conductivity were all significant parameters (Appendix A). Salinity and electrical conductivity presented high correlation (0.94), with only salinity remaining in the final model. In the final multivariate model, shown in Table 2, only TDS and salinity were significant (OR = 0.99, *p* < 0.001 and OR = 10.1, *p* < 0.05, respectively). The increase in one percentage unit of salinity is associated with a 10 times increase in the chance of finding *Leptospira* when adjusted with TDS (Table 2 and Figure 1B). In addition, the final model included two significant interaction terms: pH and type of water, and TDS and type of water. The analysis of the interaction between pH and type of water showed that sewage samples with high pH had a 25% reduction in the chance of detecting presence of *Leptospira* when compared to standing water (Table 2). Overall, an increase in one unit TDS reduced the chance of finding *Leptospira* by approximately 1%. However, when evaluating TDS in interaction with the type of water, we observed an increase of approximately 0.05% in the chance of finding *Leptospira* in sewage when compared to standing water (Table 2). In addition, Generalized Additive Model (GAM) analysis identified a positive relationship between TDS and positivity rate for the TDS interval of 15–300 mg/L and a negative relationship for the TDS interval of 300–1000 mg/L (Figure 1A).

In the univariate linear model, we found that the concentration of pathogenic *Leptospira* was associated with temperature, pH, turbidity, salinity, and electric conductivity (*p* < 0.2). The final linear model for the concentration of pathogenic *Leptospira*, however, retained only pH as a significant parameter (*p* < 0.001). Increases in pH were associated with increased concentrations of pathogenic *Leptospira* (Figure 1C). Turbidity was important in the adjustment of the final model.

## 4. Discussion

The physicochemical properties of water have been suggested to be essential for understanding the maintenance of pathogenic *Leptospira* and consequently environmental transmission. In this study, specific physicochemical characteristics were differentially related to the presence and load of leptospires identified in water samples from an urban slum. Among the parameters analyzed, TDS, salinity, and pH were associated with *Leptospira* positivity in aquatic matrices, while only pH was associated with the pathogen concentration. We also found significant physicochemical differences between sewer and standing water which affected the chances of identifying positive *Leptospira* samples.

We observed a positive trend between TDS values 0 and 300 mg/L and *Leptospira* presence. However, the lower sample size in this subgroup prevented an appropriate statistical evaluation of this trend. The interaction between TDS and type of water with *Leptospira* positivity rate also supported this trend, as observed by the higher proportion of *Leptospira* positive samples in sewage water when compared to standing water (Appendix A). Previous works have proposed that survival of pathogenic *Leptospira* in water may increase by adhesion of this bacterium to suspended soil particles or aggregation with other microorganisms [10,19]. On the other hand, we identified a negative association between TDS values superior to 300 mg/L and *Leptospira* presence rate. The reduction of *Leptospira* prescence rate may be explained by the fact that high concentrations of TDS can be lethal to aquatic organisms, causing osmotic shock [20]. This effect may affect, for exemple, *Leptospira*’s osmoregulatory strength, reflecting on the ability to adapt to the environment [21]. Besides this, the reduction of light penetration in water, a consequence of the excess of soil particles in aquatic bodies, results in reduced photosynthesis capacity in water organisms. In this scenario, a drop in the oxygen supply present in this medium is possible, thus affecting *Leptospira*’s survival, because these bacteria are obligatory aerobic [22].

We found that a 1% increase in salinity was associated with a tenfold increase in *Leptospira* positivity rate. These findings corroborate the study by Viau and Boehm (2011) [23], which observed a positive association between *Leptospira* DNA concentration and salinity rate. However, Khairani-Bejo (2004) found that the duration of *Leptospira* decreased as the salinity increased, where the organism died in seawater, where the salinity is on average 35‰. Another study showed that, even in a solution with low salt concentration (0.13% NaCl), *Leptospira*’s survival can decrease, promoting inhibition of the energetic function of this bacterium [10].

The analysis of the interaction between pH and water type showed a negative association, and sewage samples with acidic pH presented a higher chance of detecting the presence of *Leptospira* when compared to standing water. However, pH was positively associated with pathogenic *Leptospira* load. Similarly, previous laboratory studies demonstrate that *Leptospira* survival improves in alkaline environments [8,24,25]. Ideal pH value for survival is close to neutral [22] and extreme pH values are lethal for pathogenic leptospires. A previous controlled laboratory study shows that, even under identical conditions, different survival patterns among different serovars are observed. They reported that *Leptospira* serovar Icterohaemorrhagiae increased their survival time by months at pH above 7.0 [22,24]. It is known that the pH of water affects the availability of micronutrients and extreme values can have a toxic effect for some microorganisms [20].

We did not find an association between turbidity and *Leptospira* concentration. Our results are different from the ones reported by Viau and Boehm (2011) [23], which found a significant correlation between turbidity and the concentration of *Leptospira* in the river waters in Hawaii. However, the qPCR approach used in Hawaii was not specific for pathogenic *Leptospira* and detected intermediate and pathogenic species. Alternatively, the relationship between turbidity and or TDS and *Leptospira* may be related to the phenomena of precipitation, mobilization, and dilution. Pathogenic *Leptospira* are released by rodents on the soil surface [26,27], where they persist for a prolonged period [28,29,30]. During heavy rain events, pathogenic *Leptospira* can be mobilized and resuspended from the soil reservoir along with sediment, directed to open sewers and drainage with runoff and, as a result, increasing turbidity and TDS [23,31].

The average water temperature at Pau da Lima was 25 °C, and the values did not vary with the type of water. This is in agreement with a previous laboratory study that showed that a water temperature range between 25 and 30 °C allows the survival of pathogenic *Leptospira* [8]. For example, Fontaine et al. 2015 observed the greatest survival rate of *Leptospira* on days with temperatures of 20 °C and 30 °C [32]. One of the limitations of our study was that we did not measure dissolved oxygen in the water samples, which varies with temperature and can be an important parameter for *Leptospira* survival [33].

## 5. Conclusions

In conclusion, this study provides novel field data indicating a relationship between physicochemical variables (specifically TDS, salinity, and pH) and the presence and concentration of pathogenic *Leptospira* in sewage and standing water in urban slums. Further studies should aim to examine causality and confounding factors not considered in this analysis (i.e., rainfall intensity, presence of animal reservoirs, geographical diversity, etc.). As the presence of this *Leptospira* in water contributes to leptospirosis in vulnerable communities in developing tropical nations, research on the mechanisms behind *Leptospira* distribution in the environment may be valuable to inform better public health measures to reduce the burden of the disease.

## Figures and Tables

**Figure 1 tropicalmed-05-00146-f001:**
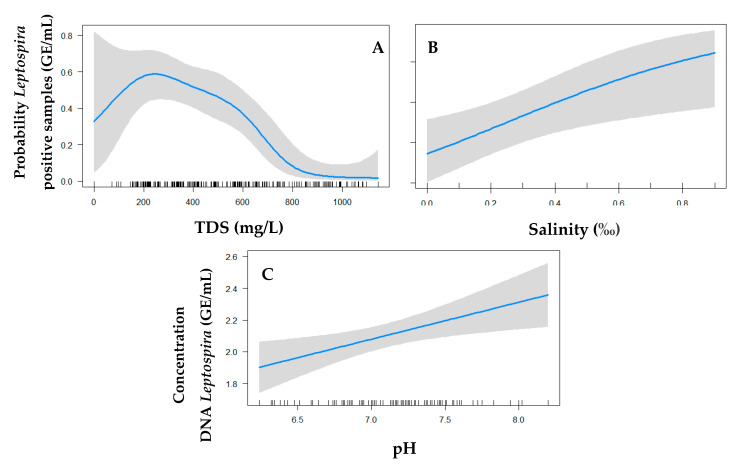
Probability of finding *Leptospira* positive water samples in the final logistic model according to TDS (**A**) and salinity (**B**). Probability of finding *Leptospira* DNA concentration water samples in the final linear model according to pH (**C**). The blue line denotes the predicted chance/probability and the grey area the 95% confidence interval. Ticks on the *X*-axis denote individual measurements.

**Table 1 tropicalmed-05-00146-t001:** Physicochemical parameters (mean, standard deviation, and *p*-value) of pathogenic *Leptospira* positive and negative samples measured in the water samples collected in Pau da Lima.

Physicochemical Parameters	Overall	Positive	Negative	*p*
Temperature (°C)	25.7 ± 2.35	25.3 ± 2.44	26 ± 2.27	<0.01
pH	7.2 ± 0.61	7.1 ± 0.60	7.3 ± 0.61	0.02
Turbidity (NTU)	325 ± 284	352 ± 292	310 ± 279	0.23
Total dissolved solids (TDS) (mg/L)	533 ± 281	436 ± 209	592 ± 302	<0.01
Salinity (‰)	0.28 ± 0.306	0.24 ± 0.35	0.30 ± 0.27	0.01
Electrical conductivity (µS)	0.99 ± 0.55	0.92 ± 0.60	1.04 ± 0.50	0.01

**Table 2 tropicalmed-05-00146-t002:** Estimated regression parameters (odds ratio and confidence interval) in the final logistic and parameters (coefficient and confidence interval) in the final linear models on the chance and probability of finding a positive sample and log10 concentration of *Leptospira*, respectively. (***) *p* < 0.01, (**) *p* < 0.05, (*) *p* < 0.1.

Physicochemical Parameters	Logistic Model		Linear Model	
	Odds Ratio	95% CI	Coefficient	95% CI
pH	1.132	0.52–2.47	0.28 **	0.09–0.48
Turbidity			0.00	−0.00–0.00
TDS	0.989 ***	0.98–0.99		
Salinity	10.097 *	1.30–91.07	−0.52	−1.34–0.30
Interaction terms				
pH * Water	0.753 **	0.61–0.93		
TDS * Water	1.005 ***	1.00–1.01		
Salinity * Water			0.37	−0.15–0.89

*p*-value *** < 0.001, ** < 0.01, * < 0.05.

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
