# Peer review of "Relationship between Physicochemical Characteristics and Pathogenic Leptospira in Urban Slum Waters"

_tropicalmed, 2020, doi:10.3390/tropicalmed5030146_

Round 1
Reviewer 1 Report
Summary
Article: Relationship between physicochemical characteristics and pathogenic Leptospira in urban slum waters. Authors: Oliveira et al., 2020
This paper aimed to evaluate the characteristics of surface waters (sewer and standing waters) at 14 sites in one of Brazil's urban slum areas. It associates the physicochemical characteristics of these surface waters with the amount and concentration of pathogenic Leptospira DNA. The contribution of this work is related to identifying environmental features that contribute to maintaining Leptospira in areas with leptospirosis where living conditions and sanitation is already compromised. From the evaluated characteristics, pH and salinity of the surface are the main factors associated with Leptospira's occurrence in this environment.
Broad comments:
One of this manuscript's strengths is the location (a slum community area) where this work was conducted. It is not easy to reach these areas to collect the samples required to evaluate the compound effect of the surface water characteristics amount and concentration of pathogenic Leptospira. The efforts of the team are commendable.
It is unclear if both the leptospira results and the physicochemical characteristics measured in this study, were collected during the same period (years), or if the team went back at a later time to collect the water samples for this physicochemical characteristics analysis (See Ln 87-89). In the methods, please clearly state if the period of sample collection for the outcome and predictor variables is different or not.
A weakness of this work relates to having aims, objectives, and hypotheses that also have methods, results, and conclusions. The inverse is also true; the methods and results should fulfill the stated objectives. An example of the observed problem in this manuscript is: the goal stated in the introduction is to look at the physicochemical characteristics of surface waters and their relation to the “presence and concentration” of Leptospira DNA. The methods outlined the types of surface waters sampled. There are results for both the separate surface waters and the combined data from all surface waters; yet, methods and statistics do not describe if the two types of surface waters were evaluated separately or if their data were combined. Later in the discussion, there is a statement in reference to “… contrary to our hypothesis, sewage samples…” (note the hypothesis for a study should be presented at then of the introduction, not in the discussion).
Specific comments
Ln 62-65: Consider updating this section to include all the objectives for which you have methods, and results. Consider including the hypotheses tested (that was presented in the discussion).
Ln 70: replace … “characterized as a urban slum community, precarious ….” with “characterized as an urban slum community, with precarious” ….
LN 71: Odd sentence “Study previously … inhabitants” consider replacing with “A study previously conducted in this neighborhood revealed high ..”
Ln 79: it is not clear what you mean by … “week periods”. Do you mean sampled one every week ? .. please clarify.
Ln 87-89: This is a confusing sentence. “The published results … in combination with the newly measured …” – it is not clear if the water samples for Leptospira were obtained at the same time (year) as the water samples used for the physicochemical properties of this work. If not, there needs to be some discussion and methods as to how the sampling gap was addressed. If it is the same water sample (collected at one point in time) was evaluated for Leptospira and the physicochemical characteristics, then the sentence needs to be clear about this.
Methods section:
Please clarify if the water samples were pooled for the evaluation of the physicochemical properties, or if the data was obtained for each water sample and then the data were pooled. Do let the reader know how was this done (criteria or reasoning to take one approach or the other).
Results:
Were there any differences in the leptospira positivity of the water sources (by water source) that is between sewage or surface waters? Please clarify. We see that there are differences in pH between sewer and surface waters. – But the relation of this finding with Leptospira's positive waters by water source is unclear, and so are the objectives that help the reader look for the methods that lead to these results.
Ln 128: there are no differences “in sewage compared to standing waters” presented in table 2. and where are the objective(s) for this comparison?
Ln 137 – 143: only 2 of the 3 panels are described in the text. Either delete one of the panels or add information in the text to address all the panels in Figure 1.
Ln 157: there are significant differences found between sewer and standing water. But please add text to clarify the objectives and the methods used – that lead to this result?
Ln 182-183: Hypothesis should be stated in the introduction. There should be methods clearly identified to address this hypothesis. We cannot introduce a new hypothesis in the discussion.
Ln 2004-2007: Check this sentence it is odd. “There is an according to previous …. “
References: All the references are showing an extra number at the beginning of each reference.
Author Response
Response to Reviewer - Relationship between Physicochemical Characteristics and Pathogenic Leptospira in Urban Slum Waters manuscript
Reviewer 1
Summary
Article: Relationship between physicochemical characteristics and pathogenic Leptospira in urban slum waters. Authors: Oliveira et al., 2020
This paper aimed to evaluate the characteristics of surface waters (sewer and standing waters) at 14 sites in one of Brazil's urban slum areas. It associates the physicochemical characteristics of these surface waters with the amount and concentration of pathogenic Leptospira DNA. The contribution of this work is related to identifying environmental features that contribute to maintaining Leptospira in areas with leptospirosis where living conditions and sanitation is already compromised. From the evaluated characteristics, pH and salinity of the surface are the main factors associated with Leptospira's occurrence in this environment.
Broad comments:
One of this manuscript's strengths is the location (a slum community area) where this work was conducted. It is not easy to reach these areas to collect the samples required to evaluate the compound effect of the surface water characteristics amount and concentration of pathogenic Leptospira. The efforts of the team are commendable.
Response: We appreciate the reviewer’s comments and suggestions. Please find below answers to all of them.
It is unclear if both the leptospira results and the physicochemical characteristics measured in this study, were collected during the same period (years), or if the team went back at a later time to collect the water samples for this physicochemical characteristics analysis (See Ln 87-89). In the methods, please clearly state if the period of sample collection for the outcome and predictor variables is different or not.
Response: We agree with the reviewer that the period of collection was not clearly stated in the manuscript. Samples for molecular quantification of Leptospira DNA results and physical and chemical characteristics were collected and measured simultaneously. We have added this details information in the methods section (see lines 79-84).
A weakness of this work relates to having aims, objectives, and hypotheses that also have methods, results, and conclusions. The inverse is also true; the methods and results should fulfill the stated objectives. An example of the observed problem in this manuscript is: the goal stated in the introduction is to look at the physicochemical characteristics of surface waters and their relation to the “presence and concentration” of Leptospira DNA. The methods outlined the types of surface waters sampled. There are results for both the separate surface waters and the combined data from all surface waters; yet, methods and statistics do not describe if the two types of surface waters were evaluated separately or if their data were combined. Later in the discussion, there is a statement in reference to “… contrary to our hypothesis, sewage samples…” (note the hypothesis for a study should be presented at then of the introduction, not in the discussion).
Response: We appreciate your comments. We evaluated two types of water: standing water and sewage. However when we evaluated the physical-chemical parameters separately we did not observe statistical differences in relation to the presence and concentration of Leptospira, so we decided to integrate the two types of water in the regression model, including the type of water as an interaction factor. For a better understanding we detail the objectives (lines 61-64) and exclude the hypotheses from the discussion.
Specific comments
Ln 62-65: Consider updating this section to include all the objectives for which you have methods, and results. Consider including the hypotheses tested (that was presented in the discussion).
Response: We agree with the reviewer's suggestion. We included in the text the objective (lines 61-64)
Ln 70: replace … “characterized as a urban slum community, precarious ….” with “characterized as an urban slum community, with precarious”
Response: We appreciate the reviewer's correction. Replacement made in the text (Please, see lines 66-67).
LN 71: Odd sentence “Study previously … inhabitants” consider replacing with “A study previously conducted in this neighborhood revealed high ..”
Response: We appreciate the reviewer's correction. Replacement made in the text (Please, see lines 68-69).
Ln 79: it is not clear what you mean by … “week periods”. Do you mean sampled one every week ? .. please clarify.
Response: We thank the reviewer for pointing this out. We collected samples for three non-consecutive days one week each month (July and January). We corrected the sentence in the text (Please, see lines 78-79).
Ln 87-89: This is a confusing sentence. “The published results … in combination with the newly measured …” – it is not clear if the water samples for Leptospira were obtained at the same time (year) as the water samples used for the physicochemical properties of this work. If not, there needs to be some discussion and methods as to how the sampling gap was addressed. If it is the same water sample (collected at one point in time) was evaluated for Leptospira and the physicochemical characteristics, then the sentence needs to be clear about this.
Response: This question was answered earlier (Please, see lines 70-72 and 87-89).
Methods section:
Please clarify if the water samples were pooled for the evaluation of the physicochemical properties, or if the data was obtained for each water sample and then the data were pooled. Do let the reader know how was this done (criteria or reasoning to take one approach or the other).
Response: Each water sample (sewage or standing water) was analyzed individually for Leptospira DNA and physical-chemical characteristics. The results obtained for each sample were grouped by type of sample for the regression analyses. We included type of water as an interaction factor in the statistical analyzes to identify whether it played a role in the probability of finding a positive sample We have included this explanation in the text (please see on lines 95-96).
Results:
Were there any differences in the leptospira positivity of the water sources (by water source) that is between sewage or surface waters? Please clarify. We see that there are differences in pH between sewer and surface waters. – But the relation of this finding with Leptospira's positive waters by water source is unclear, and so are the objectives that help the reader look for the methods that lead to these results.
Response: There was a difference in positivity for Leptospira between sewage and standing water samples, in which sewage water was more positive when compared to standing water, but we did not highlight it in the text because this data had already been reported in the previous article (Casanovas-Massana, et al (2017) 10.1016/j.watres.2017.11.068). In the present study, we observed a difference between the physical-chemical characteristics in relation to the type of water, as can be seen in table S2. Therefore, the type of water was included in the regression models as a possible interaction factor.
Ln 128: there are no differences “in sewage compared to standing waters” presented in table 2. and where are the objective(s) for this comparison?
Response: We have included an additional objective in the introduction (see lines 61-64).
Ln 137 – 143: only 2 of the 3 panels are described in the text. Either delete one of the panels or add information in the text to address all the panels in Figure 1.
Response: We thank the reviewer for this comment. We include the citation of Figure 1B in the text (Please, see line 123).
Ln 157: there are significant differences found between sewer and standing water. But please add text to clarify the objectives and the methods used – that lead to this result?
Response: We adjusted the additional objective of evaluating the physical-chemical characteristics in relation to the type of water. In the methods section, it is explained that two types of water were collected (lines 70-72), and that the parameters were averaged based on type of water and the type of water included in the logistic and linear models (lines 95-96).
Ln 182-183: Hypothesis should be stated in the introduction. There should be methods clearly identified to address this hypothesis. We cannot introduce a new hypothesis in the discussion.
Response: We thank the reviewer for this comment. We removed the hypotheses from the discussion.
Ln 2004-2007: Check this sentence it is odd. “There is an according to previous ….
Response: We modify the sentence in the text (Please, see line 203). We did a detailed review of English. All corrections made to the text are marked in yellow.
References: All the references are showing an extra number at the beginning of each reference.
Response: Thanks for noticing this mistake. We have corrected the references as suggested.

Reviewer 2 Report
Overall this is a very nice paper summarizing a study of pathogenic Leptospira in the urban slum environment. It highlights the public health importance due to the implications for disease transmission dynamics and epidemiologic risk around water sources of differing physicochemical qualities. Adequate numbers and location distribution of water samples is a key strength of this study. I wonder if the authors could discuss a little more the connection of these findings to human illness; for example, other than identifying hot spots as the authors mention, is there some education or alerting that can be done by linking pathogenic Leptospira to these areas where people are living? Also, since the authors mention that Leptospira is linked to inadequate sanitation, can they comment on the possibility of human urine shedded Leptospira (not just from the rodents) and that leading to identification of the pathogen in these urban slum waters? My last question is in regards to whether or not the authors attempted to culture any of the Leptospira from the water, as PCR positive does not necessarily mean viable infectious bacteria? Lastly, some minor editing/proofreading is required, such as line 208: "wary" should be "vary". Thank you for the opportunity to review!
Author Response
Response to Reviewer - Relationship between Physicochemical Characteristics and Pathogenic Leptospira in Urban Slum Waters manuscript
Reviewer 2
Comments and Suggestions for Authors
Overall this is a very nice paper summarizing a study of pathogenic Leptospira in the urban slum environment. It highlights the public health importance due to the implications for disease transmission dynamics and epidemiologic risk around water sources of differing physicochemical qualities. Adequate numbers and location distribution of water samples is a key strength of this study. 1) I wonder if the authors could discuss a little more the connection of these findings to human illness; for example, other than identifying hot spots as the authors mention, is there some education or alerting that can be done by linking pathogenic Leptospira to these areas where people are living? 2) Also, since the authors mention that Leptospira is linked to inadequate sanitation, can they comment on the possibility of human urine shedded Leptospira (not just from the rodents) and that leading to identification of the pathogen in these urban slum waters? 3) My last question is in regards to whether or not the authors attempted to culture any of the Leptospira from the water, as PCR positive does not necessarily mean viable infectious bacteria? 4) Lastly, some minor editing/proofreading is required, such as line 208: "wary" should be "vary". Thank you for the opportunity to review!
Response: We thank the reviewer for this positive feedback and the helpful comments, (see below for response):
1) We agree that discussing the connection between these findings and human disease is an interesting point. We have included a sentence in the text on this subject in the discussion section. (please see lines 213-216).
2) We thank the reviewer for commenting on another important point. Indeed, contamination from human urine is possible. However, given the levels of rat infestation of this urban slum and considering the fact that rats are chronic shedders, the contribution of human shedded Leptospira (only acute patients) is, we believe, negligible in the big picture of the transmission. Either way, since the focus of this manuscript is the physico-chemical characteristics of the surface waters, we thing discussing the potential animal/human reservoirs could be distracting to the scope of the paper.
3) The reviewer raises an excellent point. Although Leptospira DNA cannot directly be interpreted as viable cells, we showed in a previous study that it is a good surrogate (Casanovas-Massana, et al (2018) 10.1128/AEM.00507-18). We are preparing a further study, carried out in the same study area in order to assess the diversity of species of the genus Leptospira in water and soil samples from this slum. We have indeed been able to isolate species of pathogenic, intermediate and saprophytic Leptospira in water and soil samples.
4) We carry out the modifications as suggested. All corrections made to the text are marked in yellow.
